# Proposal of Dental Hygiene Diagnosis for Cancer Patients Based on Dental Hygiene Process of Care in Acute Care Hospitals: A Narrative Review

**DOI:** 10.3390/healthcare8030217

**Published:** 2020-07-18

**Authors:** Yuhei Matsuda, Masaaki Karino, Satoe Okuma, Kumi Ikebuchi, Mayu Takeda, Takahiro Kanno

**Affiliations:** Department of Oral and Maxillofacial Surgery, Shimane University Faculty of Medicine & Oral Care Center, Shimane University Hospital, Izumo, Shimane 693-8501, Japan; karino71@med.shimane-u.ac.jp (M.K.); okuma125@med.shimane-u.ac.jp (S.O.); ikebuchi@med.shimane-u.ac.jp (K.I.); mtakeda@med.shimane-u.ac.jp (M.T.); tkanno@med.shimane-u.ac.jp (T.K.)

**Keywords:** dental hygiene diagnosis, dental hygiene process of care, cancer, acute care hospital, NANDA-I

## Abstract

A narrative review was conducted to propose dental hygiene diagnoses for cancer patients based on dental hygiene process of care in acute care hospitals. Six researchers, including three dental hygienists, all with expertise in oral healthcare for patients with cancer, decided the review outline. All researchers reviewed the literature and developed terminology for dental hygiene diagnoses. The team then modified the terminology and discussed its clarity and acceptability to develop an initial list of dental hygiene diagnosis names according to the dental hygiene human needs conceptual model subscales. In wholesome facial image, one new diagnosis was developed. In protection from health risks, 15 new diagnoses were developed. In biologically sound and functional dentition, 10 new diagnoses were developed. In skin and mucous membrane integrity of the head and neck, 10 new diagnoses were developed. In freedom from head and neck pain, two new diagnoses were developed. In freedom from anxiety and stress, eight new diagnoses were developed. In responsibility for oral health, five new diagnoses were developed. In conceptualization and understanding, three new diagnoses were developed. Based on this study, it is necessary for the academic community to develop a better taxonomy of dental hygiene diagnoses pertaining to dental hygienist clinical practice.

## 1. Introduction

In 1993, the Dental Hygiene Human Needs Conceptual Model (DH HNCM) was theorized as a framework for advancing dental hygiene practice, education, and research [1]. Originally, the DH HNCM was derived from the Yura and Walsh Human Needs and Nursing Process Theory [2]. In particular, a series of cycles in the nursing process (assessment, diagnosis, planning, implementation, evaluation) was directly incorporated into the dental hygiene process [2]. Later, in 2000, Darby and Walsh revised the DH HNCM from 11 to 8 categories of needs (Table 1) [3]. The DH HNCM provides dental hygienists with a framework for decision making and problem solving in clinical practice and research [4]. In particular, the DH HNCM plays an important role in dental hygiene assessment and dental hygiene diagnosis in the dental hygiene process of care [5]. Dental hygiene diagnosis is defined as identifying “real or potential oral health problems and health behaviors that dental hygienists can address in their education and qualifications”. In general, the statement of dental hygiene diagnosis consists of an etiology phrase and a diagnosis phrase (signs and symptoms; Figure 1) [6].

A nursing diagnosis that encompassed Maslow’s hierarchy of needs was developed in the 1950s and 1960s. It helped prioritize and plan care based on patient-centered approaches [7]. Nursing processes have since developed separately according to different theories, including Henderson, Gordon, and Orem’s [8,9]. The North American Nursing Diagnosis Association (NANDA) defines a nursing diagnosis as a clinical judgment about responses to actual or potential health problems on the part of the patient, family, or community, and it provides nurses with an up-to-date taxonomy of nursing diagnoses (Table 2) [7]. Although the taxonomy of nursing diagnoses has some problems, it provides the basis for the selection of nursing interventions that achieve outcomes for which the nurse is accountable [10]. A nursing diagnosis statement is different from a dental hygiene process of care statement in name alone; that is, both statements consist of only one diagnosis phrase. This is also the case for diagnoses made by medical doctors or dentists. Thus, a dental hygiene diagnosis, which includes the etiology phrase, makes it difficult to classify the diagnosis and understand its contents. Therefore, it is necessary to change the rules around dental hygiene diagnoses and make each diagnosis consist of only one name.

In Japan, many dental hygienists work at acute care hospitals and play an important role in oral health management for oral cavities, or perioperative oral and dental management (POM), in cancer treatment [11]. POM was introduced into the Japanese universal health insurance system in April 2012, with the goal or ensuring good oral hygiene both prior to further treatment and in posthospitalization [11]. Cancer treatment, which involves surgery, chemotherapy, and radiotherapy, is well known for causing severe oral adverse effects with not only physical but also psychological and social aspects [11,12,13,14]. Accordingly, the job of dental hygienists who work in acute care hospitals not only includes scaling and professional mechanical tooth cleaning, but also encompasses instruction of oral mucositis management and rehabilitation of the oral function for those patients under the supervision of oral and maxillofacial surgeons and hospital dental specialists. In addition, acute care hospital dental hygienists aid in supporting the family, setting up the environment for easy management of the oral cavity, providing instructions about oral care, and cooperating with the patient’s regular general dentists for seamless oral care. Thus, the environment of acute care hospitals, where there are many cancer patients with medical, oral, dental, and nursing problems, makes it easy to develop new names for dental hygienist diagnoses. In addition, our study may play an important role in providing common sense solutions for the oral health management of patients with cancer. Therefore, the aim of this review is to develop a proposal of dental hygiene diagnoses for cancer patients based on the dental hygiene process of care in acute care hospitals.

## 2. Review Process for the Purpose of Developing Diagnoses

Revision of dental hygiene diagnoses was approached using three phases [15]. First, six members of the research team, which included three dental hygienists, all with expertise in oral healthcare for patients with cancer, gathered for a discussion to decide the review outline. Second, the team reviewed literature, and each member coined new names for dental hygiene diagnoses. Third, the team modified the names and discussed their clarity and acceptability based on two evaluation criteria—that each diagnosis name should contain as few words as possible and that the name could also be used for a nursing diagnosis. Next, an initial list of dental hygiene diagnosis names was developed for each of the DH HNCM subscales. In addition, medical, dental, and nursing conditions or clinical situations with associated dental hygiene diagnoses were presented as collaborative problems (CP).

## 3. Proposal of Dental Hygiene Diagnosis for Each Subscale

### 3.1. Wholesome Facial Image (HN1)

Wholesome facial image is defined as the need to feel satisfied with one’s own oral and facial features and breath. Facial image is determined by individuals’ perception of their physical characteristics and their interpretation of how that image is perceived by others. Facial image is influenced by normal and abnormal physical changes and by cultural and societal attitudes and values [3,16,17] (Table 3). 

For example, surgical treatment of head and neck cancer containing oral cancer frequently involves extensive resection of the facial structures, resulting in a broad range of facial disfigurements. Consequently, there is often a need to restore parts of the oral and/or facial tissues [18]. These facial disfigurements are generally associated with large impacts on patients’ quality of life, including physical and facial aesthetic changes; functional limitations in the aerodigestive tract; negative consequences on familiar, social, and work relationships; and psychosocial impacts of having cancer and facial alterations [19]. 

The diagnosis of HN1 was used for acute facial disfigurement resulting from trauma or surgery in the head and neck region.

### 3.2. Protection from Health Risks (HN2)

Protection from health risks is defined as the need to avoid medical contraindications related to dental hygiene care and to be free from harm or danger involving the integrity of the body structure and environment around the person [3,16,17]. 

Ishimaru et al. reported that preoperative oral care by a dentist and dental hygienist significantly reduced postoperative complications, such as a decrease in postoperative pneumonia and all-cause mortality, within 30 days of surgery in patients who underwent cancer surgery [20]. In addition, a multicenter retrospective investigation on the efficacy of POM in cancer patients showed that POM decreased the incidence rate of oral mucositis due to chemotherapy and/or radiotherapy [21].

POM that aims to prevent oral and systemic whole-body complications plays an important role in today’s cancer treatment. Accordingly, an HN2 diagnosis is used for when there are whole-body risk factors caused by oral condition and patient’s attitude.

### 3.3. Biologically Sound and Functional Dentition (HN3)

Biologically sound and functional dentition refers to the need for intact teeth and restorations that defend against harmful microbes, provide for adequate functioning and aesthetics, and reflect appropriate nutrition and diet [3,16,17].

This HN3 diagnosis section was developed by discussing collaborative problems with dentists because the improvement of dentition is absolutely essential in the case of dental treatments such as caries treatment and dental prosthesis. For HN3 diagnoses, dental hygienists need to cooperate with dentists to solve oral problems. 

### 3.4. Skin and Mucous Membrane Integrity of the Head and Neck (HN4)

Skin and mucous membrane integrity of the head and neck is defined as the need for an intact and functioning covering of the person’s head and neck area, including the oral mucous membranes and periodontium [3,16,17].

The cornerstones of treatment for patients with locally advanced unresectable head and neck cancers are surgery and radiotherapy, which might be delivered alone or in various combinations with chemotherapy. High-grade side effects of chemoradiotherapy for these head and neck cancers include not only oral mucositis, but also dermatitis [22].

Thus, the HN4 diagnosis was developed in anticipation of oral adverse effects from high-dose chemotherapy and radiotherapy in cancer treatment.

### 3.5. Freedom from Head and Neck Pain (HN5)

Freedom from head and neck pain is defined as being exempt from physical discomfort in the head and neck area [3,16,17].

Pain is a common symptom in cancer survivors, and about one-third of them report moderate to severe pain [23]. Pain can be acute or chronic and can be directly related to cancer, associated with diagnostic or therapeutic procedures, or an adverse effect of treatments [24]. Dental hygienists cannot directly approach the pain. Thus, the two diagnoses (acute or chronic pain) were developed to be used flexibly in situations such as when a dental hygienist instructs a patient with head and neck cancer on how to use dental rinse and softly brush teeth for relieving the pain of oral mucositis.

### 3.6. Freedom from Anxiety and Stress (HN6)

Freedom from fear and stress is defined as feeling safe and being free from emotional discomfort in the oral healthcare environment and receiving appreciation, attention, and respect from others [3,16,17].

Clinical depression as well as depressive symptoms (anxiety and powerlessness) have been reported to increase ineffective coping in cancer patients [25]. The mental condition of cancer patients may affect the behavior of oral health; unfortunately, there is no obvious evidence, but this diagnosis should be used if the patients complain about problems such as anxiety.

### 3.7. Responsibility for Oral Health (HN7)

Responsibility for oral health is defined as accountability for one’s oral health according to one’s motivation, physical and cognitive capability, and social environment [3,16,17].

The researchers reported in a previous study that there is a possibility that oral-health-related self-efficacy may change throughout the cancer treatment [26]. This is a similar situation in which oral health literacy and each readiness element may worsen. In cancer treatment, attention should be paid to not only superficial behavior but also psychological principles that affect good oral health.

### 3.8. Conceptualization and Understanding (HN8)

Conceptualization and understanding involve the need to understand ideas and abstractions to make sound judgments about one’s oral health. 

In particular, many cancer patients experience spiritual distress and disturbed conceptualization. A previous study showed that intercessory prayer was effective in religious and spiritual scores [27]. Thus, attention should be paid to understanding the patient’s whole body and mentality in addition to multiprofessional cooperation.

## 4. Limitations

Some limitations should be acknowledged. First, this is a narrative review and not a systematic review. Second, due to lack of evidence about the dental hygiene diagnosis of cancer patients, the practical tips discussed in this review rely mainly on expert opinions. Finally, the purpose of this paper was to propose diagnostic sentences by a narrative review, and it cannot therefore be immediately applied to clinical practice.

## 5. Conclusions

### 5.1. Scope for Further Research

The nursing process has so far developed based on the so-called NANDA-I, which is the creation and classification of nursing diagnosis names. In the future, dental hygienists should also create and classify dental hygiene diagnosis names based on this framework. This work will create diagnosis names and will help organize the discipline of dental hygiene. However, this work cannot be done by a single researcher; it should be led by a research team or academic society. It would be necessary for an academic society, along with focus groups, to develop dental hygiene diagnoses based on this review. The development of and agreement upon a systematic method for creating and classifying dental hygiene diagnoses will enable the advancement and subdivision of the theory of dental hygiene.

### 5.2. Clinical Relevance

This topic is relevant in today’s world, as many are affected by cancer. Furthermore, comprehensive treatment by dental hygienists in oral healthcare can be enhanced by uniform and easy-to-understand diagnosis criteria as part of the process.

## Figures and Tables

**Figure 1 healthcare-08-00217-f001:**
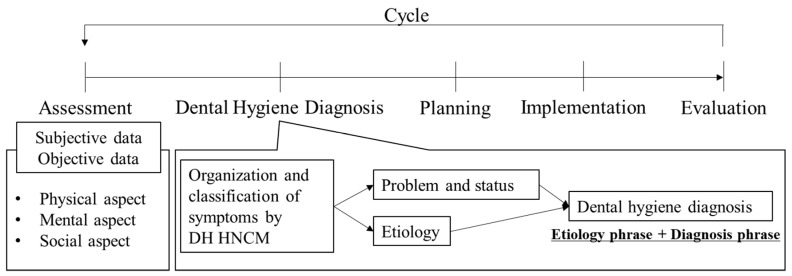
The flow of dental hygiene assessment and dental hygiene diagnosis in the dental hygiene process of care.

**Table 1 healthcare-08-00217-t001:** Dental Hygiene Human Needs Conceptual Model.

Human Needs (HN)	Contents
HN1	Wholesome facial image
HN2	Protection from health risks
HN3	Biologically sound and functional dentition
HN4	Skin and mucous membrane integrity of the head and neck
HN5	Freedom from head and neck pain
HN6	Freedom from anxiety and stress
HN7	Responsibility for oral health
HN8	Conceptualization and understanding

**Table 2 healthcare-08-00217-t002:** The North American Nursing Diagnosis Association (NANDA) taxonomy.

NANDA-I	Contents
Pattern 1	Health perception/health management pattern
Pattern 2	Nutritional/metabolic pattern
Pattern 3	Elimination pattern
Pattern 4	Activity/exercise pattern
Pattern 5	Sleep/rest pattern
Pattern 6	Cognitive/perceptual pattern
Pattern 7	Self-perception pattern
Pattern 8	Role/relationship pattern
Pattern 9	Sexuality/reproductive pattern
Pattern 10	Coping/stress tolerance pattern
Pattern 11	Value/belief pattern

**Table 3 healthcare-08-00217-t003:** Proposal of dental hygiene diagnosis categories in cancer treatment in an acute care hospital.

Human Needs (HN) Conceptual Model	Diagnosis	Defining Characteristics	Reference
HN1: Wholesome facial image	Disturbed body image	The condition in which there is confusion in regard to the mental picture of one’s physical self.	[28]
HN2: Protection from health risks	Risk for infection	The condition in which there is an increased risk of being invaded by pathogenic organisms.	[20,29,30,31]
	Delayed surgical recovery	The condition of vulnerability to an extension of the number of postoperative days required to initiate and perform activities that maintain life, health, and well-being, which may compromise health.	[32,33]
	Risk for aspiration	The condition of being at risk for entry of gastrointestinal secretions, oropharyngeal secretions, solids, or fluids into tracheobronchial passages.	[30,34]
	Ineffective self-oral health management	The condition in which patients cannot express the desire to manage oral health for treatment and prevention of complications.	-
	Ineffective family self-oral health management	The condition in which families cannot achieve the goal of oral health or are at risk.	-
	Risk for ventilator associated pneumonia	The condition of being at risk for ventilator-associated pneumonia by aspiration.	[35]
	Swallowing disorder *CP	The condition in which there has been a diagnosis of swallowing disorder. The condition in which a patient complains about deglutition.	[36]
	Feeding self-care deficit	The condition in which the ability to perform or complete feeding activities is impaired.	[37]
	Risk for oral bleeding	The condition in which there is postoperative oral bleeding or oral bleeding due to low platelet count during chemotherapy.	[38]
	Impaired physical mobility *CP	The condition in which there is a diagnosis of impaired physical mobility. The condition in which there is a limit to intentional body movements and limb movements that can be executed by patients themselves.	[39]
	Risk for ineffective respiratory function	The condition in which air passage through the respiratory tract is threatened.	[40]
	Ineffective airway clearance	The condition of inability to remove secretions or obstructions from the respiratory tract.	[40]
	Readiness for enhanced nutrition	The condition in which nutrition intake is sufficient for metabolism and further strengthening is possible.	-
	Impaired communication	The condition of inability to exchange ideas and requests with others.	-
	Impaired verbal communication	The condition in which any or all of the ability to receive, process, and convey words or letter sounds is diminished.	-
HN3: Biologically sound and functional dentition	Periodontal disease *CP	The condition in which diagnosis of periodontal disease has been made by a dentist.	[41]
	Caries *CP	The condition in which diagnosis of caries has been made by a dentist.	[42]
	Malocclusion *CP	The condition in which diagnosis of malocclusion has been made by a dentist.	[43]
	Malalignment *CP	The condition in which diagnosis of malalignment has been made by a dentist.	[44]
	Missing teeth *CP	The condition in which diagnosis of missing teeth has been made by a dentist.	[45]
	Crown fracture *CP	The condition in which diagnosis of crown fracture has been made by a dentist.	[42]
	Ill-fitting prosthesis *CP	The condition in which diagnosis of ill-fitting prosthesis has been made by a dentist.	[42]
	Peri-implantitis *CP	The condition in which diagnosis of peri-implantitis has been made by a dentist.	[46]
	Ill-fitting denture *CP	The condition in which diagnosis of ill-fitting denture has been made by a dentist.	[47]
	Masticatory disturbance *CP	The condition in which diagnosis of masticatory disturbance has been made by a dentist.	[30]
HN4: Skin and mucous membrane integrity of the head and neck	Oral mucositis *CP	The condition in which diagnosis of oral mucositis has been made by a dentist or medical doctor.	[12]
	Surgical site infection *CP	The condition in which diagnosis of surgical site infection has been made by a dentist or medical doctor.	[32,33]
	Osteonecrosis of the jaws *CP	The condition in which diagnosis of osteonecrosis of the jaws has been made by a dentist or medical doctor.	[48]
	Dry mouth	The condition in which the salivary glands in the mouth do not make enough saliva to keep the mouth wet.	[49]
	Abnormal tongue shape and color	The condition in which the tongue turns red, yellow, purple, or another color and changes shape.	-
	Stomatitis *CP	The condition in which diagnosis of stomatitis has been made by a dentist or medical doctor.	[12]
	Bite wound *CP	The condition in which diagnosis of bite wound has been made by a dentist or medical doctor.	-
	Oral burn *CP	The condition in which diagnosis of oral burn has been made by a dentist or medical doctor.	[50]
	Oral trauma *CP	The condition in which diagnosis of oral trauma has been made by a dentist or medical doctor.	[51]
	Radiodermatitis *CP	The condition in which diagnosis of radiodermatitis has been made by a dentist or medical doctor.	[52]
HN5: Freedom from head and neck pain	Acute pain	The condition in which a patient complains about acute pain.	-
	Chronic pain	The condition in which a patient complains about chronic pain.	-
HN6: Freedom from anxiety and stress	Nausea and vomiting	The condition in which a patient complains about stomach discomfort and the sensation of wanting to vomit or vomiting.	[53]
	Powerlessness	The condition of lacking self-control, which is caused by stress in patients with cancer.	[54]
	Death anxiety	The condition in which one experiences a feeling of dread, apprehension, or solicitude (i.e., anxiety) when one thinks of the process of dying or ceasing.	[54]
	Anxiety	The condition in which a patient experiences a feeling of worry, nervousness, or unease about something with an uncertain outcome.	[54]
	Fear	The condition in which a patient experiences a feeling of unpleasant emotion caused by the threat of danger, pain, or harm.	[54]
	Ineffective coping	The condition in which a patient is unable to realistically assess stressors and use available resources to cope with stress.	[55]
	Readiness for enhanced coping	The condition in which there are cognitive and behavioral efforts to manage demands that are sufficient for well-being and that can be strengthened.	[55]
	Disabled family coping	The condition in which caregiver’s experience stress when patients cannot cope with the symptoms they are experiencing.	[56]
HN7: Responsibility for oral health	Low oral-health-related self-efficacy	The condition in which the patients cannot execute courses of oral health action required to deal with prospective situations.	[26]
	Low oral literacy	The condition in which individuals have the capacity to obtain, process, and understand basic oral health information and services needed to make appropriate oral health decisions.	[57]
	Deficient knowledge	The condition in which individuals have a lack of cognitive information or psychomotor ability needed for health restoration, preservation, or health promotion.	[58]
	Readiness for enhanced knowledge	The condition in which the presence or acquisition of cognitive information related to the oral health topic is sufficient for meeting health-related goals and can be strengthened.	[58]
	Readiness for enhanced oral self-care	The condition in which a patient is able to regulate and integrate a therapeutic regimen into their daily living for the treatment of illness and its sequelae in a way that is sufficient for meeting health-related goals and that can be strengthened.	[58]
HN8: Conceptualization and understanding	Decisional conflict	The condition in which there is uncertainty about which course of action to take when the choice among competing actions involves risk, loss, or a challenge to one’s values and beliefs.	[59]
	Disturbed self-concept	The condition in which there is a negative state of change about the way a person feels, thinks, or views him or herself.	[60]
	Spiritual distress	The condition in which the ability to experience and integrate meaning and purpose in life through connectedness with self, others, art, music, literature, nature, and/or a power greater than oneself is impaired.	[58]

*CP: This problem presents medical and dental conditions or clinical situations with associated dental hygiene diagnoses and collaborative problems.

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
