# Peer review of "Proposal of Dental Hygiene Diagnosis for Cancer Patients Based on Dental Hygiene Process of Care in Acute Care Hospitals: A Narrative Review"

_healthcare, 2020, doi:10.3390/healthcare8030217_

Round 1

Reviewer 1 Report

The topic is of great interest and the structure of the paper is clear.

Please see some minor comments in the attached pdf.

Author Response

Response to Comments/Suggestions from Reviewer 1

Dear Reviewer 1:

We are truly grateful for your critical comments and thoughtful suggestions on our manuscript. We have made careful modifications accordingly. All changes made to the main text are in red. Please find our point-by-point responses to your comments/questions below.

Sincerely,

Dr. Yuhei Matsuda, Corresponding Author for the article: healthcare-848738

Comments and Suggestions for Authors

The topic is of great interest and the structure of the paper is clear.

Please see some minor comments in the attached pdf.

  1. add more details about possible connections and applications of nursing process theory in dentistry and dental hygiene fields (Page 1, Line 33)

Response: Thank you for your comment. We have added a more detailed sentence: “In particular, a series of cycles in the nursing process (assessment, diagnosis, planning, implementation, evaluation) was directly incorporated into the dental hygiene process.” (Page 1, Line 33).

  1. add figure or table which can better explain NANDA taxonomy (Page 2, Line 48)

Response: Thank you for your suggestion. We have added a table on NANDA taxonomy (Table 2; Page 2, Line 64).

  1. add more details about clarity and acceptability criteria mentioned (Page 3, line81)

Response: Thank you for your suggestion. We have added clarity and acceptability criteria: “…based on two evaluation criteria—the words in each term should be as few as possible and the term should also be used in nursing diagnosis.” (Page 3, Line 89).

Reviewer 2 Report

Despite being an interesting article, it presents a methodological error. This article is a review, but authors do not provide MeSH terms, search strategy or databases where documents were retrieved. Authors should include this information.

Author Response

Response to Comments/Suggestions from Reviewer 2

Dear Reviewer 2:

We are truly grateful for your critical comments and thoughtful suggestions on our manuscript. We have made careful modifications accordingly. All changes made to the main text are in red. Please find our point-by-point responses to your comments/questions below.

Sincerely,

Dr. Yuhei Matsuda, Corresponding Author for the article: healthcare-848738

Comments and Suggestions for Authors

Despite being an interesting article, it presents a methodological error. This article is a review, but authors do not provide MeSH terms, search strategy or databases where documents were retrieved. Authors should include this information.

Response:

Thank you for your suggestion. Our review paper does not intend to answer one clinical question. The aim is to outline the current state of dental hygiene diagnosis, and was written for the purpose of drawing a future research with reference to the development of nursing diagnosis. Therefore, we use the method classified as a narrative review, not the method classified as a systematic review or meta-analysis. Therefore, we did not provide details about the search method.

In fact, the narrative review is an established method and a scientific article applying this method has been published in the past in MDPI Healthcare. Therefore, we have used this article for reference.

Reference paper:

Fe Bozzetti, P Cotogni: Nutritional Issues in Head and Neck Cancer Patients. Healthcare (Basel). 2020 Apr 17;8(2):E102. doi: 10.3390/healthcare8020102.

However, to avoid misleading the reader, we have changed the title “Proposal of Dental Hygiene Diagnosis for Cancer Patients Based on Dental Hygiene Process of Care in Acute Care Hospitals: A Literature Review” to “Proposal of Dental Hygiene Diagnosis for Cancer Patients Based on Dental Hygiene Process of Care in Acute Care Hospitals: A Narrative Review.”

Reviewer 3 Report

Dear Authors

Thanks for the submission of the manuscript.

In general, please avoid the use of possessive pronouns (such as "we conducted..."), rather use passive voice.

The Title indicates: It is a literature review. The inclusion/exclusion criteria are not defined for the publications which were cited and summarized. Without following the PRISMA guidelines including PICO-format, risk of bias, etc. it is not a review by nature, it is more an opinion letter of an expert group.

I recommend to use the PRISMA guidelines and to restructure the manuscript: http://www.prisma-statement.org

Author Response

Response to Comments/Suggestions from Reviewer 3

Dear Reviewer 3:

We are truly grateful for your critical comments and thoughtful suggestions on our manuscript. We have made careful modifications accordingly. All changes made to the main text are in red. Please find our point-by-point responses to your comments/questions below.

Sincerely,

Dr. Yuhei Matsuda, Corresponding Author for the article: healthcare-848738

Comments and Suggestions for Authors

Thanks for the submission of the manuscript.

In general, please avoid the use of possessive pronouns (such as "we conducted..."), rather use passive voice.

The Title indicates: It is a literature review. The inclusion/exclusion criteria are not defined for the publications which were cited and summarized. Without following the PRISMA guidelines including PICO-format, risk of bias, etc. it is not a review by nature, it is more an opinion letter of an expert group.

I recommend to use the PRISMA guidelines and to restructure the manuscript: http://www.prisma-statement.org

Response:

Thank you for your comment. As per your suggestion, we have revised the use of “we” to passive voice as much as possible.

Also, our review paper does not intend to answer one clinical question. The aim is to outline the current state of dental hygiene diagnosis, and was written for the purpose of drawing a future research with reference to the development of nursing diagnosis. Therefore, we use the method classified as a narrative review, not the method classified as a systematic review or meta-analysis. Therefore, we did not provide details about the search method.

In fact, the narrative review is an established method and a scientific article applying this method has been published in the past in MDPI Healthcare. Therefore, we have used this article for reference.

Reference paper:

Fe Bozzetti, P Cotogni: Nutritional Issues in Head and Neck Cancer Patients. Healthcare (Basel). 2020 Apr 17;8(2):E102. doi: 10.3390/healthcare8020102.

However, to avoid misleading the reader, we have changed the title “Proposal of Dental Hygiene Diagnosis for Cancer Patients Based on Dental Hygiene Process of Care in Acute Care Hospitals: A Literature Review” to “Proposal of Dental Hygiene Diagnosis for Cancer Patients Based on Dental Hygiene Process of Care in Acute Care Hospitals: A Narrative Review.”

Round 2

Reviewer 2 Report

Thank you for your responses. This article can be accepted in present for

Author Response

Response to Comments/Suggestions from Reviewer 2

Dear Reviewer 2:

We are truly grateful for your critical comments and thoughtful suggestions on our manuscript. We have made careful modifications accordingly. All changes made to the main text are in red. Please find our point-by-point responses to your comments/questions below.

Sincerely,

Dr. Yuhei Matsuda, Corresponding Author for the article: healthcare-848738

Comments and Suggestions for Authors

Thank you for your responses. This article can be accepted in present for

Response:

Thank you very much for accepting our review article.

Reviewer 3 Report

Dear Authors

Thanks for the revised version of the manuscript.

Per definition the level of evidence could not support a Narrative Review compared to a Systematic Review. The change of the title helps categorize the manuscript in the correct way.

Now, you have to change the overall categorization as well: It is NOT an Article - it is a Review.

Author Response

Response to Comments/Suggestions from Reviewer 3

Dear Reviewer 3:

We are truly grateful for your critical comments and thoughtful suggestions on our manuscript. We have made careful modifications accordingly. All changes made to the main text are in red. Please find our point-by-point responses to your comments/questions below.

Sincerely,

Dr. Yuhei Matsuda, Corresponding Author for the article: healthcare-848738

Comments and Suggestions for Authors

Thanks for the revised version of the manuscript.

Per definition the level of evidence could not support a Narrative Review compared to a Systematic Review. The change of the title helps categorize the manuscript in the correct way. Now, you have to change the overall categorization as well: It is NOT an Article - it is a Review.

Response:

Thank you for your comment. According to your suggestion, we changed the overall categorization to a Review article.
